# COVID-19 Pandemic Impact on Undergraduate Nursing Students: A Cross-Sectional Study

**DOI:** 10.3390/ijerph19148347

**Published:** 2022-07-08

**Authors:** Felice Curcio, Cesar Iván Avilés González, Maria Zicchi, Gabriele Sole, Gabriele Finco, Oumaima Ez zinabi, Pedro Melo, Maura Galletta, José R. Martinez-Riera

**Affiliations:** 1Faculty of Medicine and Surgery, University of Sassari (UNISS), Viale San Pietro 43/B, 07100 Sassari, Italy; felice.curcio@aousassari.it (F.C.); mzicchi@aousassari.it (M.Z.); 2Department of Medical Sciences and Public Health, University of Cagliari, Cittadella Universitaria di Monserrato, 09042 Cagliari, Italy; gabriele.finco@unica.it (G.F.); maura.galletta@unica.it (M.G.); 3Mater Olbia Hospital, Strada Statale 125 Orientale Sarda, 07026 Olbia, Italy; gabriele.sole@materolbia.com (G.S.); oumaima.ezzinabi@materolbia.com (O.E.); 4Centre for Interdisiplinary Research in Health, Universidade Católica Portuguesa, 4169-005 Porto, Portugal; pmelo@ucp.pt; 5Department of Community Nursing, Preventive Medicine and Public Health and History of Science, University of Alicante, 03690 San Vicente del Raspeig, Spain; josera.ferranna@gmail.com

**Keywords:** anxiety, COVID-19, coronavirus disease 2019, nursing education, nursing students, online education, pandemic lockdown, psychological distress

## Abstract

COVID-19 is a challenge for education systems around the world. This study aimed to evaluate the perceived impact of the COVID-19 pandemic on nursing students, by assessing their emotions, the level of concern in contracting the virus and their perceived stress. We conducted an observational cross-sectional study. A total of 709 nursing students completed an anonymous questionnaire. The levels of anxiety and stress were assessed using the generalized anxiety disorder scale and the COVID-19 student stress questionnaire, respectively. In total, 56.8% of the sample often or always found it difficult to attend distance-learning activities. The main difficulty referred to was connection problems (75.7%). The mean generalized anxiety disorder score was 9.46 (SD = 5.4) and appeared almost homogeneous among students across the three years of study; most of the students showed mild (35%) to moderate (27%) levels of anxiety; 19% had severe anxiety. The overall COVID-19 stressor mean scores were 11.40 (SD = 6.50); the majority of the students (47.1%) showed scores indicative of moderate stress, 25% showed low stress levels, and 28% showed high-stress levels. Improvements and investments are needed to ensure high-quality distance learning, adequate connectivity, technical support for students, as well as strategies to promote mental health.

## 1. Introduction

Coronavirus Disease (COVID-19) is an infectious disease caused by a newly discovered coronavirus that was identified in Wuhan, the largest metropolitan area in China’s Hubei province, in December 2019 [1]. The common human to human transmission routes of COVID-19 are represented by direct transmission (droplet release through exhalation, coughing, or sneezing) and contact transmission (contact with contaminated surfaces and oral, nasal, and eye mucous membranes) [2].

The COVID-19 epidemic spread rapidly globally; on 11 March 2020, the World Health Organization (WHO) declared a pandemic state [3]. The first country contaminated in Europe was Italy, where the epidemic began on 21 February 2020. As of 6 February 2021, the Italian national surveillance system had reported 2,553,032 cases and 88,516 deaths from COVID-19 [4]. To limit the diffusion of the virus, the Italian government established a series of decrees aimed at containing the spread of the epidemic. In Italy, following the Italian Law of the 23 February 2020 [5,6], all Italian schools and universities suspended teaching, training, and laboratory activities, and closed all facilities such as libraries and study rooms to avoid direct contact between persons and to minimize the transmission between persons of different geographic areas. With the sudden closure, universities and colleges around the world had to transform traditional “face to face” classes into distance courses [7]. This sudden closure made it necessary to solve several problems, such as the complexity of setting up remote laboratories [8], adapting complex/traditional lessons in an e-learning format [7,9], lack of resources, difficulties in Wi-Fi connections (accessibility and broadband), as well as the lack of training among students and teachers related to e-learning platforms and their potential applications [10]. New technologies play a vital role in current education systems and the use of technology in the educational process has indeed changed learning. Students use technology as an integral part of their daily life: they carry out research, socialize, and communicate on the internet. Before the COVID-19 pandemic, several authors conducted studies on the perception of students in relation to online learning [9,11,12]. They found that the inclusion of technology in university education is advantageous [12]. However, although being connected from home (i.e., studying at home) is advantageous because students can avoid taking public transport or driving to go into university, being forced to stay at home can lead to feelings of isolation [13,14]. Cultivating social relationships to colleagues or peers is something that should not be overlooked; social relationships represent a factor of disease prevention and health improvement. University students are a special social group with active life habits based on relationships and contacts, physical and university activities, travel, and gatherings. The pandemic emergency changed their life drastically. Several studies explored the factors associated with the COVID-19 outbreak among college students. They highlighted high levels of anxiety and concern about both academic delays and the impact of the epidemic on their daily life due to the interruption of the academic routine in terms of activities, objectives, and social relationships [15,16,17]. In fact, the lockdown hindered the possibility to live an academic life, by impacting the course of study organization (e.g., delays in scheduling activities or cancellation, difficulties in using online platforms for distance learning, etc.), as well as by compromising the relationships with colleagues and professors that are crucial in students’ life [18].

In this sense, the aim of this study was to evaluate the perceived experience of the COVID-19 pandemic on nursing students, by assessing their emotions, the level of concern in contracting the virus during university activities, and their perceived stress during the pandemic lockdown.

## 2. Materials and Methods

### 2.1. Participants and Procedure

This study included an online survey involving students enrolled in the bachelor’s degree course in Nursing from two universities in the Sardinia region, Italy. University of Sassari (UNISS) and University of Cagliari (UNICA) with its branch in Nuoro city were the participating universities. The total number of students enrolled in the three years of course was 912 students.

The survey used the free-access Google Forms application in order to administer a self-reported structured questionnaire. The link to the online questionnaire was sent through the students’ WhatsApp groups. Data collection took place in about ten days, from 17 March to 27 March 2021.

### 2.2. Instrument

The survey was designed in Italian, and the questions were developed after reviewing the relevant literature and international guidelines [19,20,21,22,23]. The structured questionnaire included 28 items and consisted of four sections. The first section involved questions aimed to collecting demographic data such as age, gender, nationality, marital status, course venue, and year of study. The second section included questions regarding (1) students’ attendance to distance-learning lessons (yes, no); (2) students’ perception of distance-learning modality as a valid substitute for traditional learning (yes, no, partially); (3) students’ perceived impact of COVID-19 outbreak on both quality of course of study; and (4) professional training. These last two items were scored through a Likert-type scale ranging from 1 (not at all) to 5 (extremely). The third section detected (1) student’s worries about contracting COVID-19 during daily activities; (2) worries for their family members in contracting COVID-19; (3) perceived risk for contracting COVID-19 during both clinical placement; and (4) face-to face lessons. In addition, this section included items regarding (5) students’ perception about the risk for nurses to contracting the infection during working activity and (6) which prevention measures may protect against the infection risk. The first five items were scored using a Likert scale ranging from 1 (not at all) to 5 (extremely). The last item provided for the possibility for choosing one or more responses from a list of prevention actions (e.g., screening and phone anamnesis to identify possible critical cases, body temperature measurement, room ventilation, etc.). The last section of the questionnaire evaluated (a) the students’ psychological reactions to COVID-19 and required students to indicate which emotion in a list of five they felt when thinking about the pandemic. This scale was taken from Bellini et al.’s [24] recent study. Moreover, (2) the presence of anxiety symptoms was evaluated using seven items from generalized anxiety disorder (GAD-7) scale [25]. Students had to indicate how often symptoms of anxiety occurred in the last two weeks. Likert scale for scoring ranged from 0 (never) to 3 (almost every day). The total score of the scale ranged from 0 to 21. Scores from 0 to 4 were indicative of minimal anxiety, from 5 to 9 for mild anxiety, from 10 to 14 for moderate anxiety, and from 15 to 21 were indicative of severe anxiety. The specificity, sensitivity, and internal consistency of this scale are greater than 0.80. Finally, (3) students’ perceived stress during COVID-19 pandemic was measured using COVID-19 student stress questionnaire (CSSQ) [26]. It included 7 items on a 5-point Likert scale ranging from 0 (not at all stressful) to 4 (extremely stressful). The scale assessed three dimensions (e.g., fear of contagion, social isolation, relationship and academic life) perceived as sources of stress. The overall stress score of the scale was between 0 and 28. Scores of 6 or less indicated low levels of perceived global stress related to COVID-19, scores of 7–15 indicated moderate levels of perceived stress among students, and scores of 16 or more indicated high levels of COVID-19-related perceived global stress. The scale showed acceptable internal consistency (Cronbach’s alpha = 0.71).

### 2.3. Ethical Considerations and Data Availability Statement

According to Italian law, studies recruiting students and not including sensitive data such as critical health condition, do not require permission from the Ethical Review Authority. Faculty Directors of the Universities of Sassari and Cagliari authorized the study.

All procedures in the study were in accordance with the ethical standards of the 1964 Helsinki Declaration and its later amendments, and with the General Data Protection Regulation (EU) 2016/679 (GDPR). Informed consent was obtained from students before to complete the questionnaire. Students were informed that their participation was voluntary and anonymous, according to Italian Data protection law (e.g., Decree n. 196/2003). Additionally, students were informed that they could leave the study at any time without any adverse consequences for their university program.

### 2.4. Statistical Analyses

Data were analyzed using the SPSS (IBM, Chicago, IL, USA) version 26.0 statistical software. Descriptive statistics such as frequencies, percentages, mean and standard deviation, were performed to analyze the characteristics of the sample for the study variables. The independence chi-square (χ^2^) test was applied to compare subgroups of the sample. Frequencies and percentages for the study variables were compared between students of first, second, and third year of study to detect possible differences between groups. ANOVA analysis with post hoc multiple comparisons was performed using Bonferroni’s procedure to analyze differences between mean values of the study variables. Confidence intervals (95%) of the differences were calculated.

## 3. Results

The survey was sent to 912 nursing students. Among them, 709 completed the questionnaire (77.7%). Specifically, 61.8% (n = 438) were enrolled at the University of Sassari, 30.7% (n = 218) at the University of Cagliari, and 7.5% (n = 53) at the branch in Nuoro.

### 3.1. Demographic Information

Almost all of the students (n = 704; 99.3%) were Italian. Most of the sample were first-year students (first year n = 331, 46.7%; second year n = 218, 30.7%; third year n = 152, 21.4%) and eight (1.1%) were students who had not completed university exams within the established time period (outside the prescribed time). Most of the students (n = 654, 92.2%) were single, 50 students (7.1%) were married, and 10 students (0.7%) were divorced or separated. Almost the entire sample (n = 578, 81.5%) lived with their families, and 131 students (18.5%) lived alone.

### 3.2. Perceived Quality of Distance Learning

Almost the entirety of the sample (n = 704, 99.3%) indicated that their universities organized distance learning during COVID-19 lockdown, and 93.3% of the students (n = 664) declared that they attended the distance learning. Among them, 56.8% (n = 403) reported that they often or always encountered difficulties in attending the distance-learning activities. The main difficulty referred to was connection problems (n = 537, 75.7%), followed by the difficulty in managing applications by students or teachers (n = 466, 65.7%). The application used for distance learning was described by 58.7% (n = 416) of the participants as barely functional and 32.3% (n = 229) referred a lack of technical support. Nevertheless, to the question “Do you think distance learning can replace face-to-face teaching?”, most of the students (n = 404, 57%) answered “partially”, 25% (n = 177) answered “yes”, and 18.1% (n = 128) answered “no”.

### 3.3. Perceived Impact of COVID-19 on Study Career and Professional Future

To the question “Can the COVID-19 emergency negatively impact your education career?”, 236 students (33.3%) answered “extremely” or “a lot”, 252 (35.5%) “moderately”, and 221 students (31.1%) “little” or “not at all”.

Regarding the question “Can the COVID-19 emergency negatively impact the training for your future career?”, 39.7% (n = 281) of students answered “extremely” or “a lot”, 38.3% (n = 272) answered “moderately”, and 22% (n = 156) said that the COVID-19 emergency can influence “little” or “not at all” the training for their future career. For 261 (36.8%) students, the COVID-19 emergency will change for the worse how students will be trained for their future careers; 60 students (8.5%) said that there will be no change; and 267 (37.7%) students do not know. For 121 (17.1%) students, the COVID-19 emergency will change for the better how students will be trained.

### 3.4. Virus Contagion and Perceived Risk

Among the participants, 164 students (23.1%) had one or more relatives who contracted coronavirus, 641 students (90.4%) said that one or more acquaintances were infected, and 68 students (9.6%) declared that they did not know anyone who contracted the virus. Furthermore, 38.8% (n = 275) of responders stated to be “extremely” or “a lot” worried that their families might contract coronavirus, 41.9% (n = 297) were “moderately” worried, and only 19.3% (n = 137) were “little” or “not at all” worried for their families.

Regarding the concern to become infected during daily activities, 275 students (38.8%) indicated being “extremely” or “a lot” worried, 297 (41.9%) were “moderately” worried, and 137 (19.3%) were “little” or “not at all” worried.

Regarding clinical placement, 491 students (69.2%) reported being “extremely” or “a lot” worried that the clinical learning activities would expose them to the risk of contracting the virus, 167 (23.6%) were “moderately” worried, and 51 students (7.2%) were “little” or “not at all” worried.

Face-to-face teaching is considered for 38.2% (n = 271) of students as an activity “extremely” or “a lot” at high risk for contracting the virus, for 36.8% (n = 261) of students it is considered as “moderately” at high risk, and “little” or “not at all” at high risk for 25% (n = 177) of the students.

The perceived risk for nurses contracting the virus during their work activity is “extremely” or “a lot” probable for 69.7% (n = 494) of the participants, “moderately” probable for 25.8% (n = 183), and “little” or “or not at all” probable for 4.5% (n = 32) of the participants.

### 3.5. Prevention Measures to Reduce the Risk of COVID-19 Infection

To the question “During clinical activity, which measures do you think might prevent COVID-19 infection risk?”, the prevention strategies more frequently reported by students were “Environmental sanitation” (93.2%, n = 661), “Using individual protective equipment” (89.6%, n = 635), “Providing disinfectants and surgical masks to all patients in the waiting room” (87.4%, n = 620) “Environmental ventilation” (84.6%, n = 600), and “Reducing the number of patients in the waiting room” (70.2%, n = 498). Measures less frequently reported by students were “Body temperature measurement” (57.4%, n = 407) and “Telephonic screening/anamnesis to identify possible critical cases” (55.7%, n = 395). Other prevention strategies freely suggested by students included implementing vaccinations (2.1%, n = 15).

### 3.6. COVID-19-Pandemic-Related Emotions

Among the emotions experienced by students regarding the COVID-19 emergency, 42.2% (n = 296) of students perceived a low sense of fear, 32.1% (n = 225) moderate fear, and 25.7% (n = 180) intense fear. Concerning sadness, 46.1% (n = 323) of students perceived this emotion intensely, 31.9% (n = 224) poorly, and 23% (n = 154) moderately. With regard to the sense of worry with emergency, 45.5% (n = 319) of students indicated intense worry perception, 29.7% (n = 208) moderate worry, and 24.8% (n = 174) experienced low worry. About anxiety, 38.7% (n = 271) of participants perceived this emotion intensely, 35.8% (n = 251) poorly, and 25.5% (n = 179) felt moderate anxiety. Finally, regarding anger, 43.4% (n = 304) of students declared feeling intense anger, 23.5% (n = 165) moderate anger, and 33.1% (n = 232) experienced low perceived anger.

### 3.7. Psychological Health Outcomes

A higher percentage of students (35.09%, n = 246) indicated mild levels of general anxiety disorder, compared to 27.1% (n = 190) moderate levels, and 18.8% (n = 132) minimal levels. Nevertheless, 19% (n = 133) of students showed general anxiety disorder scores indicative of severe anxiety.

With regard to students’ stressors related to the coronavirus pandemic, the majority of the students 47.1% (n = 330) showed scores indicative of moderate stress, 25% (n = 175) showed low stress levels, and 28% (n = 196) referred high stress levels.

### 3.8. Analysis of the Differences between Student Groups

Table 1 presents descriptive characteristics of the sample including the mean and standard deviation for the studied variables by discriminating for year of study.

#### 3.8.1. Perceived Impact of Pandemic and Worries about the Contagion Risk

With regard to the perceived impact of COVID-19 on both students’ university career and their preparation for the profession, the ANOVA results and multiple comparisons from Bonferroni’s test showed that there was no significant difference between student groups (F = 1.26, *p* = 0.285; F = 0.55; *p* = 0.575, respectively) and the average values were almost homogeneous among the three years of study.

Regarding students’ worry about contracting COVID-19 infection during daily activities, the results showed a significant difference between groups (F = 10.06, *p* < 0.001). Specifically, multiple comparison revealed a difference between first-year students (M = 3.10 ± 0.98) and second- and third-year students (M = 3.39 ± 0.97; M = 3.47 ± 0.98, respectively) who showed to be more concerned than the former. There was no difference between second- and third-year students. The percentages of students who are “extremely” or “a lot” worried about contracting COVID-19 infection during daily activities were significantly higher among second- and third-year students (44% and 46.7%, respectively) than first-year students (31.7%) (χ^2^ = 15.79, df = 4, *p* < 0.01).

The same can be said for the perceived risk of infection during clinical learning (F = 19.01, *p* < 0.001). Post hoc comparisons showed a significant difference between first-year students who perceived a lower risk (M = 3.70 ± 1.04) than second- and third-year students (M = 4.12 ± 0.84; M = 4.14 ± 0.77, respectively). Additionally, in this case, there was no difference between students enrolled in years subsequent to the first academic year (second and third year). The percentage of student “extremely” or “a lot” worried significantly differed across year of study (χ^2^ = 47.68, df = 4, *p* < 0.001); the highest percentages were for second-year (79.4%) and third-year (80.9%) students, and lowest percentage was for first-year students (57.1%).

About face-to-face teaching activity as a risk of infection, the only significant difference we found was between first-year (M = 3.04 ± 0.97) and second-year students (M = 3.42 ± 1.13) who perceived a lower risk of contracting the infection than the former. There was no difference between first-year students and third-year students, as well as between second-year students and third-year students (F = 9.02, *p* < 0.001). The chi-square test supported these results (χ^2^ = 23.38, df = 4, *p* < 0.001) by showing that the percentages of students who considered face-to-face teaching as “extremely” or “a lot” risky of getting COVID-19 significantly differed across year of study. Specifically, highest percentages were observed among second-year students (50.5%) with respect to first-year (31.1%) and third-year (36.8%) students. Table 2 shows the overall results of frequency distribution of the analyzed variables for academic year.

#### 3.8.2. COVID-19 Pandemic Related-Emotions

Regarding students’ emotions about the COVID-19 emergency, ANOVA analysis showed significant differences between groups only for anxiety (F = 6.62, *p* < 0.01) and worry (F = 3.62, *p* < 0.05) emotions. About anxiety, multiple comparison revealed a difference between first-year students (M = 1.91 ± 0.85) and both second- and third-year students (M = 2.10 ± 0.87; M = 2.19 ± 0.86, respectively) who showed to be more anxious than the former. There was no difference between second- and third-year students. As emerged from the chi-square test, the percentages of students who perceived high levels of anxiety significantly changed across years of study (χ^2^ = 14.23, df = 4, *p* < 0.01); percentages were higher for second-year (42.7%) and third-year (48%) students than for first-year (31.7%) students.

As regards worry, the only one significant difference we found was between first-year (M = 2.12 ± 0.83) and third-year students (M = 2.32 ± 0.78), who perceived higher worry than the former. There was no difference between first- and second-year students and between second- and third- year students. However, this result is not supported by the chi-square analysis that revealed no significant difference between groups (χ^2^ = 7.30, df = 4, *p* = 0.12), although the percentage of students who perceived high worry about the pandemic was higher among second-year (48.2%) and third-year (51.3%) students than first-year (41.1%). Regarding the emotions such as fear, sadness, and anger, no significant difference was found between groups for both mean values and frequency distribution of the variables. Table 3 shows the overall results of frequency distribution of the analyzed emotions for academic year.

#### 3.8.3. Psychological Health Outcomes

With regard to the general anxiety disorder (GAD) scores, the ANOVA results showed nonsignificant differences between groups (F = 0.83, *p* = 0.44). In fact, the mean score of GAD appears almost homogeneous among students across the three years of study (M = 9.24 ± 5.62, for first-year students; M = 9.84 ± 4.99, for second-year students; M = 9.39 ± 5.52, for third-year students). The overall mean score for GAD was 9.46 (SD = 5.41) and all the student groups reported scores referring to moderate levels of GAD. The chi-square test supported the ANOVA results by showing uniformity in percentages of students with severe general anxiety across year of study (18.4% for first-year, 20.2% for second-year, 18.4% for third-year students) (χ^2^ = 7.30, df = 4, *p* = 0.12).

Regarding students’ stressors related to the coronavirus pandemic, ANOVA analysis highlighted that there were significant differences between groups (F = 9.23, *p* < 0.001). Specifically, multiple comparison revealed a difference between first-year (M = 10.30 ± 6.52) and both second-year and third-year students (M = 12.22 ± 6.21; M = 12.60 ± 6.51, respectively) who perceived higher levels of COVID-19-related stressors than the former. There was no difference between second- and third- year students. The overall mean score for students’ stressors was 11.40 (SD = 6.50) and all the student groups reported scores referring to moderate levels of perceived stressors. The chi-square test supported the ANOVA results (χ^2^ = 23.06, df = 4, *p* < 0.001) by showing that the frequency distribution of students with high COVID-19-related stressors significantly differed across year of study with higher percentages for second- and third-year students (23% for first-year, 28.9% for second-year, 37.5% for third-year students). Table 4 shows the overall results of frequency distribution of the psychological health outcomes for academic year.

## 4. Discussion

Since the start of the COVID-19 pandemic, several surveys have been carried out to assess the impact of this emergency on healthcare professionals [19,21,27,28,29,30], and special attention has been drawn to the psychological impact. Some recent studies report data about the impact that the emergency of SARS-CoV-2 is having on the academic education of future nurses [31,32,33] and show that the pandemic contributes to increasing students’ negative emotions about the profession, increased anxiety levels, and unwillingness to practice their profession in the future. However, to our knowledge there is no study existing in Italy which assessed the psychological experience of the pandemic for nursing students.

Most of the governments around the world temporarily closed educational institutions to contain the spread of the COVID-19 pandemic. Such a strategy was a part of a physical-distancing policy to slow the transmission of the infection and ease the burden on healthcare systems. In fact, the results of our study show that all the universities in the Sardinia region promptly activated distance-learning activity, and 94% of the students declared that they attended this modality of teaching. However, students considered this teaching methodology as just a “partial substitute” of traditional teaching (57%), probably because of the “technical” difficulties encountered “many times or always” by more than half of the students (403, 56.8%), and due to the lack of social interactions as well. On the one hand, the perception about the poor quality of distance learning finds support from Oducado and Soriano’s study [34], by showing ambivalence and negative attitudes of students towards e-learning during the COVID-19 pandemic. Similarly, a study conducted in India during the lockdown reported unfavorable attitudes by nursing students towards online classes [35]. On the other hand, our study contrasts with the results obtained by Luo, Geng, Pei, Chen, and Zou [36], which emphasize the positive results of distance learning on the performance, learning efficacy, and satisfaction among final-year nursing students in the city of Wuhan. In this learning program, synchronization techniques and simulation software were implemented to facilitate interactive learning among students and increase access to feedback.

At the time of the study, students were not attending clinical or internship activities. This could be the reason why the negative impact of COVID-19 on the preparation for the future profession was rated from moderately to extremely high by the majority of the students. ANOVA results showed that there was no significant difference between student groups and the average values for the variable were almost homogeneous among the three years of study. A similar situation is reflected in the results from Ilankoon et al.’s study [32] that shows a gap between teaching and learning, disruption of academic calendars, cancellation of clinical placements, disruption of professional development, and the inability of students to carry out adequate clinical activities.

About 40% of the students indicate that the COVID-19 emergency will change for the worse the way they are trained for their future profession. This is an important percentage, but it is considerably lower than that emerged from Bellini et al.’s study among dental hygiene students (53.9%) [24]. Such a different perception could depend on the period in which the survey was performed. In effect, Bellini and colleagues conducted their study immediately after the suspension of face-to-face teaching (April/May 2020). Our study was carried out approximately one year after the lockdown, when clinical learning and practice activities were restarted in an online form at least. In this sense, students have probably gained a greater awareness about the COVID-19 emergency’s impact on their future careers.

Regarding students’ knowledge about preventive measures to contrast the infection, the results showed a good knowledge of the recent guidelines concerning COVID-19 prevention procedures. This result is in line with Clements’s study [20]. Moreover, our study shows better responses for personal protective equipment, disinfection, and sanitization procedures if compared to prevention measures applied to patients. Similar results were found in previous surveys conducted on other student targets in Israel and Italy [24,37,38]. This can be due to the probability that students are more concerned about protecting themselves than their patients.

With regard to both clinical learning and frontal teaching, our students reported a high risk of getting coronavirus during these activities; nearly the entirety of the participants were either “extremely”/”a lot” or “moderately” concerned about contracting COVID-19 infection.

As regards clinical training activities, first-year students perceived a lower risk for contracting coronavirus than second- and third-year students. It is probable that students enrolled in the years following the first have reached higher awareness of the practices to be performed on patients during the internship. In addition, at the time of the survey, first-year students had not yet started the clinical activity and probably they did not have a real perception of the risk they were exposed to.

The most common feelings experienced by students at high levels are sadness, worry, and anger; fear and anxiety are instead experienced at mild to moderate levels. Similar results were found among Turkish nursing students [39], where concern was found to be an important stressor. Interestingly, while emotions such as sadness, anger, and fear are felt in a similar way among the students across the three years of the course, the levels of anxiety and worry are higher for third-year students who probably fear that their course of study could be delayed or that their clinical practice will not provide adequate training for the profession. Similar results also emerged from Bellini et al.’s study [24].

Another relevant result involves general anxiety disorder for which severe anxiety is perceived by 19% of the students. This result is higher than that revealed in previous research [22].

Regarding psychological health and emotions related to the pandemic, we found that irritability, fear of contagion, problems with relaxation, and depressed mood were the central aspects that emerged also among nursing students in China [40] and maintained the structure of the anxiety–depression network of Chinese students during the pandemic.

Regarding the students’ stressors related to the pandemic, the majority of the students showed scores indicating moderate and severe stress. The high prevalence of stress among university students was also supported in recent research [41]. Moreover, in line with Mashaal et al. [42] we found differences in perceived stressors related to distance learning between first-year students and both second- and third-year students.

This study presents some limitations that should be discussed. First, although we recruited a copious sample of nursing students, we used a convenience sample from one Italian region, which reduces the possibility of generalizing the results to all Italian students. Second, the study sample mostly included first-year students compared to second- and third years students. First-year students typically are in the process of adjusting their learning styles and habits and transitioning from high school to university; this could potentially have affected the results of our study. Third, the data were collected using a self-administrated questionnaire, thereby rating bias might be present. However, all the variables analyzed in the study refer to students’ perception about their experiences during the pandemic. In this way, collecting data using self-reported measures is adequate for perception data.

Despite the limitations, our findings provide an overview of the status of undergraduate nursing students after the COVID-19 lockdown and suggest possible intervention strategies to protect their well-being. The universities will need to capitalize on this experience and reorganize their e-teaching and e-learning approaches. When social distancing is required for public health issues, the peculiar moments of the traineeships need to be replaced by virtual reality systems. In addition, learning programs including synchronization techniques and simulation software should be implemented to facilitate interactive learning among students and increase access to feedback. It is essential that teachers who are not familiar with technology be trained in using online and virtual learning environments. The nursing course staff should stay in continuous contact with the students beyond online teaching, by identifying and supporting their needs. Moreover, it is important to maintain a stable educational framework that includes reducing to minimum any changes in the teaching schedule, announcing information about changes as soon as possible, and supplying updated information about the continuance of the academic activities and exams [22]. In addition to these strategies, problem-solving skills and coping with unavoidable stress by relating to anxiety in a healthy way (e.g., relaxation or mindfulness) [43] should be implemented to promote psychological resilience among university students. Sustainable development [44] underlines the importance of new learning methods to promote self-care and personal knowledge for good mental health. It would be useful to design or tailor university-based interventions (e.g., psychoeducation, emotional self-regulation, and positive mental health promotion) to address students’ biopsychosocial needs [45]. In addition, techniques to increase self-esteem, the perception of self-control, and self-efficacy can strengthen control over their clinical practices. Finally, from the student feedback here, improvements and investments are required to assure adequate internet connection technology, to train teachers to use e-learning platforms, and to provide technical support to students.

## 5. Conclusions

The results of the present study indicate that nursing students at the studied universities perceive the impact of COVID-19 on their study career as high. Students indicated a high level of stress related to the suspension of all clinical training activities, fear of contracting COVID-19, and dealing with the challenges of distance education. University has an important role in developing a sense of control and providing a stable educational structure for the students.

## Figures and Tables

**Table 1 ijerph-19-08347-t001:** Descriptive characteristics of the sample for all the study variables.

			N	Mean	SD	F (*p* Value)
Perceived negative impact of COVID-19 on university career		first year	331	2.99	1.152	1.256 (0.285)
		second year	218	2.98	1.056	
		third year	152	3.14	1.032	
		Total	701	3.02	1.098	
Perceived negative impact of COVID-19 on preparing students as nursing professionals		first year	331	3.21	1.112	0.555 (0.575)
		second year	218	3.30	1.043	
		third year	152	3.22	1.062	
		Total	701	3.24	1.080	
Worry about contracting COVID-19 infection during daily activities		first year	331	3.10	0.977	10.063 (<0.001)
		second year	218	3.39	0.974	
		third year	152	3.47	0.983	
		Total	701	3.27	0.990	
Clinical training activities as a risk of infection		first year	331	3.70	1.041	19.014 (<0.001)
		second year	218	4.12	0.836	
		third year	152	4.14	0.767	
		Total	701	3.93	0.949	
Face-to-face teaching activity as a risk of infection		first year	331	3.04	0.973	9.017 (<0.001)
		second year	218	3.42	1.134	
		third year	152	3.16	1.019	
		Total	701	3.19	1.047	
COVID-19-pandemic-related emotions	Fear	first year	331	1.77	0.809	2.635 (0.72)
		second year	218	1.84	0.794	
		third year	152	1.95	0.817	
		Total	701	1.83	0.808	
	Anxiety	first year	331	1.91	0.848	6.618 (0.001)
		second year	218	2.10	0.867	
		third year	152	2.19	0.859	
		Total	701	2.03	0.863	
	Worry	first year	331	2.12	0.827	3.624 (0.027)
		second year	218	2.25	0.806	
		third year	152	2.32	0.777	
		Total	701	2.21	0.813	
	Sadness	first year	331	2.14	0.888	0.993 (0.371)
		second year	218	2.09	0.868	
		third year	152	2.22	0.845	
	Anger	first year	331	2.03	0.865	2.126 (0.120)
		second year	218	2.15	0.851	
		third year	152	2.19	0.897	
GAD7		first year	331	9.24	5.62	0.831 (0,436)
		second year	218	9.84	4.989	
		third year	152	9.39	5.521	
		Total	701	9.46	5.411	
CSSQ		first year	331	10.31	6.523	9.232 (<0.001)
		second year	218	12.21	6.219	
		third year	152	12.61	6.507	
		Total	701	11.40	6.502	

**Table 2 ijerph-19-08347-t002:** Frequency distribution for Academic year about the perceived risk to contract COVID-19.

		Academic Year					
		First Year		Second Year		Third Year	
		N	%	N	%	N	%
Worry about contracting COVID-19 infection during daily activities	“Little” or “Not at all”	78	23.6%	36	16.5%	21	13.8%
	“Moderately”	148	44.7%	86	39.4%	60	39.5%
	“Extremely” or “A lot”	105	31.7%	96	44.0%	71	46.7%
Chi-square = 15.794; DF = 4; *p* = 0.03							
Clinical learning activities as a risk of infection	“Little” or “Not at all”	40	12.1%	8	3.7%	3	2.0%
	“Moderately”	102	30.8%	37	17.0%	26	17.1%
	“Extremely” or “A lot”	189	57.1%	173	79.4%	123	80.9%
Chi-square = 47.678; DF = 4; *p* < 0.001							
Face-to-face teaching activity as a risk of infection	“Little” or “Not at all”	93	28.1%	49	22.5%	33	21.7%
	“Moderately”	135	40.8%	59	27.1%	63	41.4%
	“Extremely” or “A lot”	103	31.1%	110	50.5%	56	36.8%
Chi-square = 23.379; DF = 4; *p* < 0.001							

Note. Little or Not at all ≤2.00; Moderately 2.01–3.00; Extremely or A lot ≥ 3.01.

**Table 3 ijerph-19-08347-t003:** Frequency distribution for Academic year about the perceived emotions.

		Academic Year					
		First Year		Second Year		Third Year	
		N	%	N	%	N	%
Fear	Low perception	154	46.5%	88	40.4%	54	35.5%
	Moderate perception	98	29.6%	76	34.9%	51	33.6%
	High perception	79	23.9%	54	24.8%	47	30.9%
Chi-square = 6.622; DF = 4; *p* = 0.157							
Anxiety	Low perception	135	40.8%	72	33.0%	44	28.9%
	Moderate perception	91	27.5%	53	24.3%	35	23.0%
	High perception	105	31.7%	93	42.7%	73	48.0%
Chi-square = 14.231; DF = 4; *p* = 0.007							
Worry	Low perception	95	28.7%	50	22.9%	29	19.1%
	Moderate perception	100	30.2%	63	28.9%	45	29.6%
	High perception	136	41.1%	105	48.2%	78	51.3%
Chi-square = 7.301; DF = 4; *p* = 0.121							
Sadness	Low perception	110	33.2%	73	33.5%	41	27.0%
	Moderate perception	64	19.3%	53	24.3%	37	24.3%
	High perception	157	47.4%	92	42.2%	74	48.7%
Chi-square = 4.555; DF = 4; *p* = 0.336							
Anger	Low perception	118	35.6%	65	29.8%	49	32.2%
	Moderate perception	84	25.4%	56	25.7%	25	16.4%
	High perception	129	39.0%	97	44.5%	78	51.3%
Chi-square = 9.298; DF = 4; *p* = 0.054							

Note. Low perception ≤ 2.00; Moderate perception 2.01–3.00; High perception ≥ 3.01.

**Table 4 ijerph-19-08347-t004:** Frequency distribution for Academic year about the psychological health outcomes.

		Academic Year					
		First Year		Second Year		Third Year	
		N	%	N	%	N	%
GAD7 *	minimal anxiety	71	21.5%	30	13.8%	31	20.4%
	mild anxiety	111	33.5%	80	36.7%	55	36.2%
	moderate anxiety	88	26.6%	64	29.4%	38	25.0%
	severe anxiety	61	18.4%	44	20.2%	28	18.4%
Chi-square = 5.753; DF = 6; *p* = 0.451							
		**Academic year**					
		**first year**		**second year**		**third year**	
		**N**	**%**	**N**	**%**	**N**	**%**
CSSQ **	low stress	103	31.1%	37	17.0%	35	23.0%
	moderate stress	152	45.9%	118	54.1%	60	39.5%
	high stress	76	23.0%	63	28.9%	57	37.5%
Chi-square = 23.065; DF = 4; *p* < 0.001							

Note. * minimal anxiety 0–4; mild anxiety 5–9; moderate anxiety 10–14; severe anxiety 15–21. ** low stress ≤ 6; moderate stress 7–15; high stress ≥ 16.

## Data Availability

The datasets used and/or analyzed during the present study are available from the corresponding author on reasonable request.

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
