# Peer review of "COVID-19 Pandemic Impact on Undergraduate Nursing Students: A Cross-Sectional Study"

_ijerph, 2022, doi:10.3390/ijerph19148347_

Round 1

Reviewer 1 Report

Abstract line 20: do not start a sentence with a number. A total of 709 nursing students.

Abstract lines 21-22, the levels of anxiety and stress were assessed using the Generalized Anxiety Disorder scale and the COVID-19 Student Stree Questionnaire respectively. 

2.1 Participants and procedure, the psychometric properties of the questionnaires were not reported. The authors may want to report properties such as internal consistency, content validity, etc. 

Results section, line 156, 22.3% of the students didn't complete the survey. Were they different from those who completed the survey? The authors may want to report the characteristics of those who didn't participate in the survey.

Results section, line 162, did the study investigators invite more first-year students than 2nd and 3rd years? First-year students typically are in the process of adjusting their learning styles and habits and transitioning from high school to college. The high percentage of participants being freshmen could potentially impact the study results. 

Tables 1, 3, & 4:  the authors may want to incorporate F and p values into the table so that the readers can see which student groups were compared. 

Reviewer 2 Report

i suggest that these authors review the writing and the clarity of the manuscript; also, they overlook the psychological impact 

and I suggest to review their quantitative data and link them in the discussion t strong and relevant recommendations

Reviewer 3 Report

Dear authors, the manuscript needs a grammatical review of the English language, either by a native speaker, or by a specialist/professional in the field.

In the introduction, in particular, the grammatical issue needs to be reviewed. I observe that by not using a standardized and/or validated instrument, but a questionnaire prepared by the authors, the study may have less validity regarding the analyzed data, I suggest that you propose a validation of the instrument.

Round 2

Reviewer 1 Report

The authors have addressed most of my concerns. 

Reviewer 3 Report

Dear authors. After the requested considerations, I see the study with potential publication. Sincerely.